# Material-Sparing Feasibility Screening for Hot Melt Extrusion

**DOI:** 10.3390/pharmaceutics16010076

**Published:** 2024-01-05

**Authors:** Amanda Pluntze, Scott Beecher, Maria Anderson, Dillon Wright, Deanna Mudie

**Affiliations:** Global Research and Development, Small Molecules, Lonza, 64550 Research Road, Bend, OR 97703, USAdeanna.mudie@lonza.com (D.M.)

**Keywords:** hot melt extrusion, feasibility, material sparing, copovidone

## Abstract

Hot melt extrusion (HME) offers a high-throughput process to manufacture amorphous solid dispersions. A variety of experimental and model-based approaches exist to predict API solubility in polymer melts, but these methods are typically aimed at determining the thermodynamic solubility and do not take into account kinetics of dissolution or the associated degradation of the API during thermal processing, both of which are critical considerations in generating a successful amorphous solid dispersion by HME. This work aims to develop a material-sparing approach for screening manufacturability of a given pharmaceutical API by HME using physically relevant time, temperature, and shear. Piroxicam, ritonavir, and phenytoin were used as model APIs with PVP VA64 as the dispersion polymer. We present a screening flowchart, aided by a simple custom device, that allows rapid formulation screening to predict both achievable API loadings and expected degradation from an HME process. This method has good correlation to processing with a micro compounder, a common HME screening industry standard, but only requires 200 mg of API or less.

## 1. Introduction

Many active pharmaceutical ingredients (APIs) display poor solubility, which causes challenges for their oral delivery [1,2]. One option to overcome this is by molecularly dispersing the API into a polymer matrix to create an amorphous solid dispersion (ASD). These dispersions have potential to display increased solubility and dissolution rate due to the higher free energy of the amorphous state [3]. Hot melt extrusion (HME) is one method for producing ASDs, where the polymer and API are processed under elevated temperature and shear at typical residence times of 1–10 min, dependent on the extruder setup and material properties [4,5]. HME offers a high-throughput manufacturing route without the need for organic solvents, but can pose significant challenges for thermally labile drugs and those that exhibit low polymer solubility [4].

To assess the feasibility of HME for a given compound, it must be determined whether acceptable manufacturability, API loading, performance, and chemical and physical stability can be achieved. Focusing on manufacturability, the primary consideration is whether the API can be solubilized in the intended polymer to the desired loading while maintaining acceptable purity. For best accuracy, this should be evaluated under conditions physically relevant to the HME process (time, temperature, and shear), as both API dissolution and degradation are kinetic processes. The API should be used in its ingoing form as factors such as polymorphism and crystal size can impact the overall solubility, dissolution and degradation kinetics. Lastly, since API availability is often limited at the earliest stages of development, feasibility screening is ideally material-sparing, using sub-gram quantities if possible.

Significant progress has been made towards screening, formulation development, and manufacturing scale-up for HME [4,6,7,8,9,10]. Despite these advances, many of the early-stage, small-scale techniques are not physically relevant to the extrusion process, and tend to predict thermodynamic solubility of an API in polymer without considering dissolution kinetics or degradation.

Models are often the first approach for predicting achievable API loading, using structural and/or experimentally derived inputs to predict API solubility in polymers [6,11,12,13,14,15]. While model-based approaches provide insight into formulation possibilities, they are aimed at predicting the thermodynamic solubility. This may be a good way to rank-order polymer selection, but is not necessarily a loading that can be achieved during extrusion if the dissolution of the API crystals is rate-limiting. Degradation is not accounted for either.

Of the various analytical approaches, one of the most commonly reported means of predicting API solubility in polymers at various temperatures is with differential scanning calorimetry (DSC) [14,16,17,18,19,20,21,22,23,24]. Applying thermal treatments to physical mixtures of API and polymer, the API solubility can be determined by measuring the resulting melting-point depression [20,22,23], glass transition temperature (T_g_), or residual crystalline melt enthalpy [18,19,20]. Alternatively, supersaturated ASDs are held isothermally and observed for recrystallization or phase separation after quenching [20,21]. While DSC methods are material-sparing, their use may be restricted by APIs that degrade before or during their melt, and T_g_-based measurements can be challenging for systems where API and polymer have similar T_g_ values. Also, results can vary depending on the specific method used [16]. Multiple physical mixture preparations and experimental iterations are generally required. Additionally, no mixing is employed and long experimental times on the order of hours may be required to reach equilibrium, which can compete with degradation, potentially altering the results.

Thermal microscopy has been employed to visually monitor an API’s dissolution into molten polymer [12,14,24,25,26,27]. This technique can yield useful insights into the impacts of API properties such as size, morphology, and even crystalline defects [25]. However, like DSC, this method lacks application of mixing. Additionally, the polymer must be visibly transparent and API crystals large enough to display birefringence. This method can suffer from sample biasing resulting from the relatively small viewing window.

Rheology is used to predict appropriate extrusion temperatures and API loadings, with the added benefit of providing insight for extrusion processability by monitoring viscosity [24,27,28,29,30,31,32,33,34]. In general, complex viscosity is monitored on physical mixtures of varying API loadings subjected to a constant shear rate across a range of temperatures or vice versa, where the change in viscosity indicates miscibility of the system. However, for comparison between samples, testing must be performed in their linear viscoelastic region [31]. Additionally, extrapolation of the data to the high shear stresses experienced in an extruder is needed for oscillatory rheology, requiring various methods to ensure the samples obey the Cox–Merz rule [35] for confidence in the extrapolation. Rheology-based analyses require multiple experimental iterations to build the dataset, as well as mixtures prepared at various loadings. Like the other techniques discussed, degradation is not evaluated during this testing.

Film casting is used to approximate API solubility in polymer, where solutions of varying drug load are prepared and dried into films and subsequently assessed for crystallinity [36,37]. While it is a relatively straightforward test, film casting has many drawbacks. For example, a common solvent for the API and polymer is required, drying needs to be employed to remove residual solvent, and the results can be dependent on the solvent used [38]. Most notably, it is not physically relevant of extrusion.

In addition to techniques for assessing API solubility in polymer, small-scale devices have been developed to generate the end-product extrudate-like ASD material for performance and stability characterization. These devices are not meant for predicting the API solubility in polymer, and would not be indicative of achievable API loading during large-scale extrusion due to the required pretreatment of the API. For example, the miniaturized extruder prototype mimics the final stage of the extrusion process by forcing molten material through a die, but requires pretreatment of the formulation with solvents [39]. Vacuum compression molding creates a homogeneous solid disk of controlled geometry after compression and heating of samples under vacuum, but materials must be pretreated for size reduction or solvent-casted [40,41]. Further, neither of these devices incorporates mixing or physically relevant timescales, and would therefore not be good for assessing degradation.

The most physically representative approach for determining HME feasibility is to use small-scale extruders, often employed for formulation and process development [9,12,26,37,39,42,43]. Since these are simply small-scale versions of the commercial-scale extruders, the screw design, temperature zones, and feed rate can all be adjusted. While great for studying and optimizing the extrusion processing window for scale-up, they are not necessarily ideal as a starting point when the temperature processing range and achievable API loadings have yet to be defined. For this reason, micro compounders are commonly used for HME feasibility screening [17,30,44,45,46,47,48,49]. They have conical screws that send material into a recirculating chamber, allowing time, temperature and shear (via rpm) to be studied as independent variables. Results from analysis of the resultant processed material can help to design the proper extrusion processing window. Even though the process only requires 2–15 g of mixture per run (depending on the instrument model), that can quickly lead to substantial material burden. Also, when using the recirculation chamber to evaluate the impacts of residence time, substantial yield loss is incurred due to material holdup.

These HME screening approaches are summarized in Table 1, which highlights the fact that none are able to meet all of the ideal requirements for feasibility screening: an HME process-relevant approach to determine achievable API loading and expected degradation using sub-gram quantities of API.

Thus, the goal of this work was to develop a practical material-sparing approach to determine the feasibility of manufacturing an ASD by HME. A screening procedure using more physically relevant conditions (time, temperature, and mixing) compared to the existing methods was targeted to better predict achievable API loading and expected degradation, while using only milligram quantities of API. Piroxicam (PXCM), ritonavir (RTV), and phenytoin (PHY) were used as model compounds with copovidone (PVP-VA64) as the dispersion polymer to develop experimental protocols. A flowchart for executing HME feasibility screening was developed, which results in good predictions for both achievable API loading and expected degradation for a given extrusion condition.

## 2. Materials and Methods

### 2.1. Materials

PXCM and PHY were purchased from TCI Chemicals (Tokyo, Japan) and Spectrum Chemical (New Brunswick, NJ, USA), respectively. RTV was provided by Abbott Laboratories (now AbbVie Inc., North Chicago, IL, USA). PVP-VA64 (T_g_ = 108 °C) was purchased from BASF (Ludwigshafen, Germany). All model compounds are poorly soluble drugs as per the biopharmaceutics classification system (BCS) [50], as detailed in Table 2. HPLC-grade methanol (MeOH) and acetonitrile (ACN) were purchased from Honeywell (Charlotte, NC, USA). Tri-fluoro acetic acid (TFA) and KH_2_PO_4_ were purchased from Fisher (Hampton, NH, USA). All materials were used as received. All water (H_2_O) used was MilliQ grade with 18.2 MΩ resistance.

Physical mixtures were prepared by weighing an appropriate amount of each component into a scintillation vial and mixed with acoustic mixing at 50 G for 2 min using a LabRAM II (Resodyn Corporation, Butte, MT, USA).

### 2.2. Degradation Prescreening

Degradation prescreening is the first step in the HME screening workflow, where degradation-based temperature limits are determined by two separate methods: thermogravimetric analysis (TGA) and visual assessment. These analyses, performed on the pure components along with a 50/50 physical mixture of the two, define the maximum processing temperature, which is the average of the gravimetrically and visually determined limiting temperatures (described in subsequent sections).

#### 2.2.1. Gravimetric Degradation Assessment

A small amount (ca. 3–5 mg) of sample was placed into a pre-tared aluminum pan and subjected to a series of 15 min isothermal holds from 120 up to 240 °C in 20 °C increments using a Discovery TGA (TA instruments, New Castle, DE). The percentage mass loss was recorded at each isothermal step. For each sample, the limiting temperature is defined as the lowest temperature at which 0.5% ≤ mass loss < 1%. If no single data point matches this criterion, then the average of the highest temperature at which the mass loss is <0.5% and the lowest temperature at which mass loss ≥ 1% is used. The degradation-limiting component is the sample (API, polymer, or mixture) that results in the lowest limiting temperature out of the three.

#### 2.2.2. Visual Degradation Assessment

A small amount (ca. 5–10 mg) of sample was placed into an aluminum pan for 15 min in an oven equilibrated at each temperature from 120 °C up to 220 °C in 20 °C increments. The sample is then removed and allowed to cool. All final samples are arranged in order of increasing temperature. The limiting temperature is the temperature at which an onset of color change from the initial sample is noted. If a drastic jump in color is noted, rather than an onset of color change, the limiting temperature is defined as the average of the highest temperature showing no visible color changes and the temperature where the color change is observed. The degradation-limiting component is the sample (API, polymer, or mixture) that results in the lowest limiting temperature out of the three.

### 2.3. Feasibility Screening

The second step of the HME screening workflow is to determine feasibility of HME: Can HME result in an ASD of desired API loading while maintaining acceptable purity? This is accomplished by processing physical mixtures of API and polymer in a custom apparatus, termed the MiniMixer, that allows the user to apply stirring for a desired amount of time at a given temperature. Processed samples are subsequently analyzed by powder X-ray diffraction (PXRD) and high-performance liquid chromatography (HPLC) to determine the amorphous API content and API purity, respectively.

#### 2.3.1. The MiniMixer Device

The MiniMixer (Figure 1) is manufactured with stainless steel and consists of two components: the chamber and the stirring rod. The chamber seats inside the well of an aluminum heating block (Chemglass Arex-6 Digital Pro with 28 × 98 mm aluminum block topper for 40 mL scintillation vials), allowing the device to be equilibrated to a given temperature. It is a single piece containing the sample cavity in the center where the physical mixture to be processed is loaded. It is a cylindrical cavity 5 mm in diameter and 37 mm in length, with a 45° bevel at the base. The stirring rod is 4 mm in diameter, with a 45° beveled tip. The stirring rod is controlled by a Ryobi HJP003 12 V power drill with the trigger fully depressed, allowing the stirring rod to spin at 520 rpm. The rotational speed was measured using a tachometer (model 461995, Extech Instruments, Nashua, NH, USA) in non-contact mode, with a small piece of reflective tape adhered to the stirring rod for laser detection.

#### 2.3.2. Processing Procedure

When using the MiniMixer to predict the HME achievable API loading at a particular temperature and residence time (quantified by PXRD as the amorphous content in the MiniMixer processed sample), the API loading in the physical mixture is chosen based on the processing temperature relative to the API’s melt. A physical mixture likely to lead to residual crystals is desired, without having an abundance of undissolved solids, which can lead to difficulties mixing in the device and poor sample recovery. Therefore, as a starting point, a physical mixture containing 25 wt% API is used if the API melt is >20 °C above the processing temperature. Otherwise, a physical mixture with 50 wt% API is used, under the assumption that significantly higher loadings will be possible as the melt temperature is approached. Higher loadings can subsequently be analyzed if these initial trials lead to fully amorphous processed material.

The following procedure details how a sample is processed with the MiniMixer.

The entire device is equilibrated to the desired temperature.Physical mixture (roughly 200 mg) is loaded into the sample cavity.Spinning of the stirring rod is initiated as it is fully into the sample cavity.Stirring is maintained for the desired time, in 30 s intervals with a 10 s delay between each.At the completion of mixing, the direction of the drill is reversed while the stirring rod is removed, which will be encased by the bulk of the processed sample. This is immediately placed into a liquid nitrogen-filled mortar.A razor blade is used to remove the sample from the stirring rod. Any sample remaining in the cavity is scraped out using a hooked spatula and added to the liquid nitrogen.After the liquid nitrogen evaporates, the sample is hand-ground with a mortar and pestle, quickly recovered, and allowed to equilibrate to room temperature prior to analysis.

### 2.4. Micro Compounder Extrusion Processing

Lab-scale extrusion was performed with the Haake MiniLab 3 micro compounder (Thermo Fisher Scientific, Waltham, MA, USA). The micro compounder (illustrated in Figure 2) has a 7 mL temperature-controlled chamber that houses a set of conical co-rotating screws 109.5 mm in length. Unprocessed material is fed through a port near the back end of the screws, which then convey the material through the backflow channel if the bypass valve is closed (recirculation mode) or out the extrusion channel if the bypass valve is open (extrusion mode).

Once equilibrated to the desired temperature, approximately 6 g of physical mixture was manually added through the inlet funnel in roughly 2 g increments. The total time for complete addition of material was approximately 3 min. Once all material was added, the samples were run in recirculation mode for 3 min at 200 rpm, before being extruded for collection. Material was extruded through the 4 × 1 mm rectangular opening without an additional die. Extrudates were either ground with a tube mill at 13,000 rpm or manually with a 1Zpresso Z-pro hand mill.

### 2.5. Summary of Processed Samples

#### 2.5.1. MiniMixer and Micro Compounder Trials

Table 3 summarizes the materials prepared by the MiniMixer and micro compounder for this study, which were run in two processing rounds (3 min of mixing was applied for all). A single sample from each condition was made, except for the round 1 MiniMixer samples, which were prepared in duplicate to test the reproducibility of the MiniMixer processing (see Appendix A).

The first round of processing was focused on testing API loading. First, samples were processed with the MiniMixer at the maximum processing temperature in order to determine the achievable API loading. Then, micro compounder processing was performed with a range of API loadings, in order to assess the accuracy of the MiniMixer loading predictions.

The second round of processing was conducted in order to determine the purity over a range of temperatures and compare results between the two processing methods. The temperatures used in this round are the gravimetrically and visually determined limiting temperatures (see Section 3.1). These used a single API loading, one determined to be amorphous from round 1 (see Section 3.3).

#### 2.5.2. Unmixed Isothermal Holds

Unmixed samples were prepared to be more representative of the traditional analytical screening approaches detailed in Table 1, where mixtures are thermally treated without applied mixing. These were generated to compare to the MiniMixer processed samples in order to determine whether applying a more HME-relevant process results in improved accuracy of API loading and degradation predictions. Small aluminum-weight boats were filled with physical mixtures, then placed into an oven equilibrated to the desired temperature. At specified time points, samples were removed and allowed to equilibrate to room temperature, then hand-ground with a mortar and pestle for analysis. A separate pan was used for each time point, with two replicates per time point. Samples were held isothermally at their maximum processing temperature, as summarized in Table 4.

### 2.6. PXRD

PXRD was utilized to assess crystalline content of MiniMixer and micro compounder samples using a MiniFlex 600 X-ray diffractometer (Rigaku Corporation, Tokyo, Japan) equipped with a copper anode (Kα1 = 1.5406 Å; Kα2 = 1.5444 Å) generator at 40 kV and 15 mV and a D/teX ultrahigh-speed detector. Samples were loaded onto 0.2 mm-deep Si (510) zero-background cups and scanned over 3–40° 2θ at a rate of 2.5°/min continuous scanning mode.

Quantification of amorphous API (*API_am_*, assumed to be solubilized) was determined using the ratio of the peak heights in the samples relative to the ingoing physical mixture, according to the following equations:$$API_{Xtal}\;\left(\%\right)=API_{PM}\times \left(\frac{h_{HME}}{h_{PM}}\right)$$
$$API_{am}\;\left(\%\right)=API_{PM}-API_{Xtal}$$
where *API_Xtal_* is the undissolved API, *API_PM_* is the wt% of the API in the ingoing physical mixture, and *h_HME_* and *h_PM_* are the peaks heights for an API peak at a specific 2θ value in the processed sample and ingoing physical mixture, respectively. The predicted API loading that can therefore be achieved to generate an ASD from an HME process is then determined by:$$Achievable\;API\;loading\;\left(wt\%\right)=\frac{API_{am}}{\left(API_{am}+Poly_{PM}\right)}\times 100$$
where *Poly_PM_* is the wt% of polymer in the ingoing physical mixture.

Peak fitting was performed in SmartLab Studio II software (version 4.2.132.0). The above quantification was performed on multiple characteristic peaks for each API system. The results from each were averaged for the final reported value. The PXCM samples used the crystalline peaks at 2θ values of 8.6, 11.6, 12.4, 14.5, 17.7, 21.7, and 27.3. The PHY samples used 11.3, 12.9, 16.5, 17.2, 18.1, 20.3, 22.4, and 27.7. No processed RTV samples showed residual crystallinity, and therefore no quantification was performed.

### 2.7. Purity

Purity was determined by HPLC analysis. Samples were quantified against API standards prepared at the same concentration, with purity calculated as percentage area of the API peak with respect to the total area of all detected peaks. Two replicates were prepared for each sample and standard, with one unique sample per processing condition.

#### 2.7.1. Ritonavir

An Agilent 1100 HPLC equipped with a DAD was used with an Agilent Eclipse Plus C18 column (4.6 × 150 mm, 3.5 μm particle size) held at 25 °C. Samples were dissolved in MeOH at an RTV concentration of 1 mg/mL, with 3 µL injected for analysis. An 8 min isocratic method at 1 mL/min flow was run using 0.1% TFA in 55/45 (*v*/*v*) ACN/H_2_O, with a detection wavelength of 254 nm.

#### 2.7.2. Piroxicam

An Agilent 1100 HPLC equipped with a DAD was used with an Agilent Eclipse Plus C18 column (4.6 × 150 mm, 3.5 μm particle size) held at 25 °C. Samples were dissolved in MeOH at a PXCM concentration of 0.3 mg/mL, with 3 μL injected for analysis. An 8 min isocratic method at 1 mL/min flow was run using 0.1% TFA in 60/40 (*v*/*v*) ACN/H_2_O, with a detection wavelength of 230 nm.

#### 2.7.3. Phenytoin

An Agilent 1200 HPLC equipped with a DAD was used with a Waters XBridge C18 column (4.6 × 150 mm, 5 µm particle size) held at 25 °C. Samples were dissolved in MeOH at a PHY concentration of 0.6 mg/mL, with 10 µL injected for analysis. A 45 min gradient method at 1 mL/min was run using an aqueous solution of 50 mM KH_2_PO_4_ pH 3.50 (mobile phase A) and MeOH (mobile phase B), with a detection wavelength of 229 nm. The gradient applied is summarized in Table 5.

## 3. Results

### 3.1. Maximum Processing Temperature

The TGA degradation prescreening results are shown on the left of Figure 3, where the wt% mass loss for the polymer, API, and 50/50 physical mixtures are plotted against temperature. The dashed lines show the 0.5 and 1 wt% thresholds. The colored data highlight those defining the limiting temperature. Without additional interactions between the API and polymer, the mass loss of the mixture at a given temperature would be expected to fall between that of the pure components, as is the case for the RTV system. The gravimetrically determined temperature limit from this analysis is 140 °C, limited by the API. However, as is the case for both PXCM and PHY, the mass loss of the mixture is greater than that of either pure component, suggesting the API degrades at a lower temperature in the presence of the polymer and/or vice versa. This is especially evident for the PXCM system, where the mixture shows 2.9% mass loss at 180 °C, but only 0.5 and 0.07 for the API and PVP-VA64, respectively. The limiting temperature for the PHY system was determined to be 230 °C, based on both the 220 and 240 °C data points. For the PXCM system, it was set as 170 °C, based on both the 160 and 180 °C data.

These trends are mirrored in the visual analysis (Figure 3, right). For the RTV system, it is clear that the API is the limiting component when comparing the appearance across all temperatures. Obvious color change in the API and mixture is first seen at 160 °C, with little noticeable difference between the two. However, at the higher temperatures RTVs observed, browning is most severe, with the mixture showing a hue in between that of the two components. For the PXCM system, the mixture clearly defines the temperature limitation, like the TGA suggested. The mixture appears bright yellow at 140 °C and looks completely burnt at 180 °C, but no evidence of color change is noted for the API or polymer until 180 and 200 °C, respectively. The pure API never exhibits the bright-yellow color seen in the mixture, just a slight tanning effect at 180 °C. In the mixture, the color difference between 120 °C and 140 °C is quite stark, and thus the onset of color change is said to be 130 °C, the visually determined limiting temperature. The mixture is also limiting for the PHY system, with onset of color change observed at 200 °C. While the differences are less obvious than for the other systems, the mixture is deemed to be limiting, since it appears similar in color to the pure polymer rather than an “average color” of the pure components.

The final maximum processing temperature for each system is the average of the limiting temperatures from each prescreening experiment, as summarized in Table 6.

### 3.2. MiniMixer: Small-Scale HME Screening Device

To achieve the goal of running a more HME process-relevant screening test with sub-gram quantities of API, a simple device, termed the MiniMixer, was designed and manufactured in-house (not commercially available). The entire device is heated to the desired temperature, and mixing time is easily controlled by the duration of the stirring rod’s rotation, allowing for mixing at both HME-relevant time and temperature. The stirring rod dimensions were chosen such that the resultant gap between the cavity wall and stirring rod is 0.5 mm, slightly larger than the 0.1–0.3 mm overflight gap of typical extruders [9]. With the current setup, the stirring rod spins at 520 rpm, resulting in the material experiencing a shear rate of 218 s^−1^ according to the following equation:$$Shear\;rate\;\left(s^{-1}\right)=\frac{\;\pi \times D\times n}{h\times 60}$$
where *D* is the outer diameter of the stirring rod, *n* is the rpm, and *h* is the gap width. This shear rate is of a similar order of magnitude to the average shear rates material would experience in a micro compounder [56] or extruder [57]. This allows for the MiniMixer to provide physically relevant mixing. It should be noted, however, that a distribution of shear rates occurs in both micro compounders and extruders, with the maximum shear (experienced in the overflight gap) being significantly higher than a materials’ experienced shear in the MiniMixer, especially as the extruder scale and screw speeds are increased (see Appendix A).

Beyond the numerical value of shear, in an extruder the various screw elements assist with the mixing and homogenization of the material. The stirring rod of the MiniMixer, however, was designed as a straight shaft for simplicity of material recovery. The bulk of the sample becomes wrapped around the stirring rod and is easily recovered. Remaining material in the sample cavity is removed with a hooked spatula.

### 3.3. Achievable API Loadings

Based on degradation prescreening results, physical mixtures were processed on the MiniMixer at 150 °C for PXCM and RTV and 215 °C for PHY. PXCM and PHY used 25 wt% physical mixtures, while a 50% mixture was used for RTV (based on the criteria outlined in Section 2.3.2). Processing of the 25% PHY mixture resulted in no detectable crystalline content, and therefore a second mixture with 35 wt% PHY was processed to quantify the degree of solubilized API. The resulting achievable API loadings were found to be ≥50, 22.1, and 29.0 wt% for RTV, PXCM, and PHY, respectively. No residual crystals were identified in the RTV sample, which is why the loading is said to be ≥50 wt%: higher drug loadings were not tested. These values are called the achievable API loading because this is the amount of API able to be solubilized into the polymer at a given temperature and time via HME, which may not necessarily be the thermodynamic solubility.

To test these predictions, physical mixtures of various API loadings were prepared and processed on the micro compounder. Figure 4 shows their resultant diffractograms. Micro compounder results were in good agreement with the MiniMixer predictions. For PXCM, a 20 wt% sample appears fully amorphous with no residual crystalline peaks evident, whereas both the 25 and 30 wt% samples show crystallinity. Only one 50 wt% API loading was run for RTV, which appears fully amorphous. For PHY, the 35 wt% sample shows residual crystals, while the 25 and 30 wt% loadings are amorphous. The solubilized API content was quantified by PXRD for the non-amorphous micro compounder samples, as summarized in Table 7.

It should be noted that samples appearing amorphous by PXRD may have residual crystals below the limit of detection. PXRD is generally accepted to have limits of detection of a few wt% [58]. However, for the purposes of feasibility screening, amorphous PXRD is sufficient to define a successful ASD formulation, as the purpose is to estimate the achievable API loading with minimal material and time, in order to decide as early and quickly as possible if HME is a viable route to pursue.

Overall, the results show good correlation with the initial API loading estimates, indicating good predictability of the MiniMixer.

To compare the MiniMixer screening approach to the more typical analytical methods that do not employ mixing (see Table 1), physical mixtures of RTV (50%), PXCM (25 wt%), and PHY (35 wt%) were held isothermally over the course of 4 h, determining the extent of API dissolution by PXRD. All time points for the RTV unmixed samples were fully amorphous as expected, since the isothermal hold was processed above its melting temperature and it is not a rapid crystallizer [52]. The PXCM and PHY results are detailed in Figure 5. At 215 °C, PHY reached its maximum loading of ~30 wt% within 15 min, as evidenced by no further increase in solubilized API up to an hour. This loading shows good agreement to what was achieved with both the MiniMixer and micro compounder. PXCM, on the other hand, requires greater than 2 h to reach >20 wt% solubilized API compared to the >20 wt% loadings achieved in 3 min with the MiniMixer and micro compounder. While the unmixed sample reaches the same extent of solubilized API after 4 h, the data over time do not appear to level off, as was observed with PHY, indicating this is not the thermodynamic solubility.

### 3.4. Degradation

To test how well degradation correlates from the MiniMixer with the micro compounder, samples were prepared by both methods at three temperatures: the maximum processing temperature and both the gravimetric and visually determined limiting temperatures (see Table 6). Figure 6 summarizes the purity results from these processed samples for each API. Additionally, degradation of the unmixed isothermal holds was analyzed as a representation of the typical analytical approaches that do not employ mixing (see Table 1) to compare to the MiniMixer approach, and are summarized in Figure 7. Table 8 shows the comparisons of the MiniMixer processed and unmixed samples (3 min predicted values) with the micro compounder.

When comparing the MiniMixer and micro compounder results, similar shapes in the degradation profiles are observed for both PXCM and RTV. In both cases, the MiniMixer shows less degradation compared to the micro compounder, but with good agreement (less than 2% difference) between the two processing techniques, except PXCM at 170 °C, which demonstrates a 6.3% difference between the MiniMixer and micro compounder. This is above the determined maximum processing temperature of 150 °C, however. In both cases, the onset of significant API degradation aligns well with the suggested maximum processing temperature determined from the degradation prescreening.

Without mixing, both systems show similar degradation kinetics of 0.032% and 0.038% purity loss per minute for RTV and PXCM, respectively (as determined by a linear fit to the Figure 7 data). The MiniMixer and micro compounder processed samples align well with the unmixed sample degradation kinetics for RTV, yet higher deviation is observed for the PXCM processed samples. For PXCM, both MiniMixer and micro compounder samples show greater extents of degradation than the isothermal hold sample after 3 min. In order to reach the same degradation that occurred during micro compounder processing, an unmixed PXCM sample would need to be held for 59 min, whereas only 7 min would be required for the unmixed RTV sample.

For PHY, minimal API degradation is observed for all MiniMixer and micro compounder processed samples, even at 230 °C. The unmixed PHY samples exhibit 0.003% purity loss per minute, which would require an unmixed sample to be held for 9 min to match the observed API degradation of the micro compounder processed sample.

## 4. Discussion

### 4.1. Degradation Prescreening

The degradation prescreening serves as a quick and easy way to establish the preliminary processing temperature range for MiniMixer processing, without relying on analytically burdensome techniques (e.g., HPLC).

Knowing the processing temperature range resulting in acceptable levels of degradation within the expected HME residence time is important during early feasibility screening, yet often not investigated beyond TGA studies. TGA is a commonly employed method to determine HME processing temperature limits, since it is easy to execute and requires little material [59]. However, use of TGA as the sole method of degradation screening has drawbacks, since it only detects readily volatile degradants, risking overestimating safe temperatures. For example, Surasarang et al. found that up to 90% of albendazole had degraded during extrusion despite processing at a safe temperature determined by TGA [13]. In addition, volatile loss not corresponding to degradation, such as solvent loss and sublimation, can convolute the results. Due to the potential for misleading results using TGA alone, visual analysis was chosen as a second technique for degradation prescreening.

Visual assessment relies on color change, which is generally associated with degradation and may be a product-critical quality attribute. However, color change on its own is not a reliable indicator of maximum processing temperature. For example, the degree of color change will be dependent on the specific material and amount of degradation, may not be readily obvious, and is subjective to the observer. This is exemplified by the set of data presented herein, where the PXCM color change is drastic, but subtle in the other systems. Additionally, degradation is not necessarily associated with color change at all. This is demonstrated by the RTV data set, where no noticeable difference in color was noted on the 160 °C micro compounder sample (see Appendix A), despite significant API degradation (0.6% impurities). Conversely, color change is not necessarily indicative of degradation. For example, the initial yellowing observed on the PXCM mixture in this study (Figure 3) may be attributable to simple proton transfer [60].

Although use of TGA or visual assessment on its own may have a higher probability of misleading results, the two techniques used in combination can lead to more accurate prediction of the appropriate maximum processing temperature.

Utilizing this combination of methods for pure PVP-VA64 suggests a maximum processing temperature of 215 °C (Table 6), in good agreement with the manufacturer’s claim of 220 °C [61]. The pure polymer is included to ensure no processing beyond its limits. Considering polymer degradation is important, not only to ensure it remains within the compendial limits of the material but also because polymer degradation can alter the ASD performance [62,63].

Another key consideration for prescreening is that the polymer/API mixture is included instead of relying solely on the API, since excipient interactions and/or drug dissolution itself (where the API is no longer in its crystalline form, resulting in faster degradation kinetics) can reduce the degradation onset temperature [13,64,65]. This is especially prominent in the PXCM system, where the maximum temperature would have been set as 185 °C based on the API, yet the mixture dictates the limit to be 150 °C (Table 6), which still resulted in high levels of degradation on the actual extrudate (95.8% purity). In fact, of the 37 unique API/polymer combinations we have screened to-date (which include 6 polymers), 49% show degradation limited by the mixture rather than either of the pure components.

Polymer degradation is likely occurring in the PHY system and responsible for the mass loss and color changes noted in the mixture during degradation prescreening. This seems reasonable, as the mixture’s results in both prescreening tests (i.e., the mass loss noted by TGA and the observed color changes) are more similar to those from the pure PVP-VA64 than those of the pure API, which does not show any evidence of degradation on its own until at least 240 °C. This hypothesis is further supported by observations during purity analysis: the 1, 2, and 4 h unmixed samples did not fully dissolve into the diluent (even after 24 h in solution), despite complete extraction of PHY. All samples for the other two API systems readily dissolved. Additionally, while no significant API degradation occurred, slight discoloration was observed on the 215 and 230 °C micro compounder extrudates (see Appendix A). While the authors acknowledge that being mindful of polymer degradation is important, quantification of polymer-specific degradation was not investigated beyond the degradation prescreening.

Incorporating a degradation prescreening step is a helpful starting point to minimize MiniMixer experimental iteration and material usage. However, as no mixing or mechanical stress is applied and API-specific degradation is not evaluated, selection of HME temperatures using *only* these results is likely to incorrectly predict degradation under extrusion conditions, as demonstrated by analysis of unmixed samples herein. Additionally, degradation can be complex when considering the shear and other stresses induced by the extrusion process [13,46,65,66]. This is where the value of the feasibility screening using the MiniMixer comes into play, as it generates a sample emulating the HME conditions that can be analyzed for both API degradation and loading.

### 4.2. MiniMixer Predictability

The MiniMixer allows for a sample to be processed at HME process-relevant temperatures, times, and shear rates. This applied mixing provides a sample comparable to an extrudate from a micro compounder, but with an order of magnitude less API.

Along with the applied mixing, a primary advantage of this technique is its universal application to all APIs. Of the three systems studied herein, RTV is the only one for which loading could be easily studied with melt-based assessments like those often employed in DSC (for example, melting-point depression). It is not surprising that high loadings of RTV were possible, as the mixtures were processed above its melting temperature and it is not a rapid crystallizer [52]. Similar predictions would have thusly been expected from an unmixed sample. On the other hand, PXCM and PHY were processed well below their melting points, and determining the API loading using these more traditional approaches would be challenging because PXCM degrades at its melting point [67] and the PHY melt is significantly higher than the PVP-VA64 degradation.

Rather than an assessment that relies on melting, another option would be to hold an unmixed sample isothermally at temperature for a given time, return it to ambient conditions, and then determine the solubilized portion through a given measurement (e.g., the change in glass transition temperature upon rescanning).

In practice, because the solubilization time is unknown and thermodynamic solubility is targeted, mixtures are not exposed to temperature for the same amount of time as the expected HME residence time for any of the traditional analytical approaches (i.e., DSC, thermal microscopy, and rheology), often being exposed to temperature for hours. However, allowing a mixture to equilibrate to solubility for these long experimental times leads to degradation. All three systems show significant API degradation over the course of 4 h. If one were to select a hold time to match the HME residence time, only the RTV results would have led to accurate predictions of both achievable API loading and expected degradation.

In the case of PHY, only degradation can be accurately predicted for an unmixed sample held for the 3 min micro compounder residence time. Figure 5 shows that after 5 min at 215 °C, the predicted achievable API loading is 2.5 wt% lower (26.5 wt%) than from 3 min of MiniMixer processing (29.0 wt%). Assuming the 30.8 wt% loading calculated from the 35 wt% extrudate is the true achievable loading (see Table 7), the 5 min unmixed leads to 14% error (a 3 min unmixed hold would be expected to show a greater discrepancy), whereas the MiniMixer loading prediction is a 5.8% error. While an unmixed sample of PHY reaches its apparent solubility within a relatively short timeframe of 15 min, one may have expected PHY to show longer API dissolution kinetics compared to PXCM, based on the difference between the melting points of the API and the hold temperatures (~80 °C and 50 °C difference for PHY and PXCM, respectively). Additional factors, such as relative API particle sizes and/or the viscosity differences, may also explain the varying dissolution kinetics between these two systems. Either way, leaving this mixture at temperature for hours not only leads to API degradation, but in this case polymer degradation as well, both of which can lead to changes in the T_g_ (PVP-VA64 exhibits a 3 °C T_g_ increase after 4 h exposure to 215 °C; see Appendix A), again making interpretation by DSC methods complicated.

Applying an exact residence time match to PXCM would have resulted in the worst predictions for both loading and degradation. Due to its slow dissolution rate, PXCM mixture held without mixing at 150 °C for the same extrusion time of 3 min would have only resulted in <10 wt% solubilized API, even though it was possible to achieve a nearly 25% ASD via HME in the micro compounder. The degradation would also have been underpredicted, despite having the same degradation kinetics as the RTV sample, which does match well with the degradation of the extrudate. Alternatively, degradation would be grossly overestimated if allowed to reach equilibrium for accurate solubility results, requiring the sample to be held for longer than 4 h to reach the same API loading possible on the micro compounder. Utilizing the MiniMixer with the same extrusion time yields good correlation for both key factors, and would have led to a recommendation of a processing temperature below 150 °C due to the degradation. This is in agreement to what has been demonstrated by Schlindwein et al. [68]. They studied extrusion of PXCM/PVP-VA64 on a Leistritz Nano16 small-scale extruder, and reached similar findings utilizing an iterative DOE process: 20 wt% PXCM loading and a maximum set temperature of 140 °C.

The collective results of these studies highlight the importance of physically relevant mixing (enabled by the MiniMixer processing), which subsequently allows for physically relevant timescales. Without this mixing, the timescales required to reach the achievable API loading and resultant degradation can vary widely depending on the API–polymer system of interest, evidenced by the three systems studied herein. The overall chemical miscibility, along with API crystal size, relative free energy of its polymorphic form, polymer viscosity and subsequent plasticization by the API, can all impact relative dissolution kinetics. There would be no easy way of defining the “right” timescale for an unmixed sample beforehand. The combination of the physically relevant mixing and time at temperature achieved with the MiniMixer allows for robust predictions of both API loading and expected degradation, regardless of formulation.

While MiniMixer processing is meant to provide a reasonably good estimate of what is possible with extrusion, it is not intended to replace lab-scale extrusion equipment or provide exact predictions.

While overall more accurate than unmixed samples, the results are slightly lower than what occurred in the micro compounder, particularly for degradation. This is not surprising, as the feed time to get all material into the micro compounder (before mixing begins) is approximately 2–3 min, resulting in extended thermal exposure times for a subset of the material. Additionally, the higher peak shear rate in the micro compounder may cause local temperature increases caused by viscous dissipation [10], whereas no significant temperature spikes were measured in the MiniMixer (see Appendix A). This increased temperature could increase the achievable API loading in the micro compounder compared to the MiniMixer. Similarly, both processing methods may underestimate achievable API loading compared to large-scale extruders that have higher peak shear rates and the addition of screw elements.

There is also potential for the drug loading of the MiniMixer’s ingoing physical mixture to influence the amount of solubilized API within the processing time, much like a solution slurry will reach thermodynamic solubility faster with higher concentrations of undissolved solids (up to ~100-fold excess) that help push towards equilibrium [69]. Evidence for this can be seen in the PXCM micro compounder samples processed from 25 and 30 wt% physical mixtures (Table 7). The amount of amorphous API calculated from the 30% sample is 27.6 wt%, despite the 25 wt% extrudate clearly containing PXCM crystals (achievable loading of 24.4 wt%), suggesting a faster rate of dissolution for the higher API-loaded physical mixture.

### 4.3. HME Screening Procedure

Based on the testing conducted in this work, a flowchart for executing HME feasibility screening was established, as shown in Figure 8. The overall process for each step is described below for binary API–polymer systems, applicable to all extrudable polymers (see Appendix A for additional data using PHY processed with hydroxypropyl methylcellulose, HPMCAS). It should be noted that multicomponent formulations, such as those containing plasticizers, are able to be processed with the MiniMixer and analyzed in a similar manner, but efficient incorporation of plasticizers has not yet been fully explored. Further details describing the rationale for selection of the specific parameters are provided in the Appendix A.

#### 4.3.1. Degradation Prescreening

This step defines the processing temperature window for the feasibility screening. As a starting point, the minimum processing temperature for a given API–polymer system is presumed to be 20 °C above the glass transition temperature of the intended polymer (108 °C for PVP-VA64, see Appendix A), and the degradation prescreening is used to approximate the maximum processing temperature, the highest temperature one can expect to process at without causing significant degradation to the API or polymer.

The pure components and 50:50 physical mixtures are subjected to 15 min isothermal holds from 120 °C up to 240 °C and analyzed by both gravimetric and visual analyses. The ease, speed, and universality of these techniques (i.e., not API dependent methods) make them ideal to quickly estimate processing limits. The two techniques are used cooperatively, rather than relying on a single method, as each individual technique has its own drawbacks.

#### 4.3.2. Feasibility Screening

The goal of the feasibility screening is to determine whether HME is a viable manufacturing route to make an ASD for a given formulation by predicting the possible API loading and expected degradation relating to a set temperature and time. These results guide the decision for if and how to proceed with HME, based on what is desired for the ASD to achieve the required metrics (e.g., performance and dose, dosage form mass/size, specified purity, etc.).

The first MiniMixer run is performed at the maximum processing temperature determined during degradation prescreening, and subsequently analyzed by both PXRD and HPLC to assess API loading and degradation, respectively. This is to minimize the number of experimental iterations: if the desired API loading is not possible at this temperature, going higher to reach the loading is not recommended because of degradation. A single physical mixture (~200 mg) is used for this first screening: 25 wt% API if its melt temperature is >20 °C above the processing temperature, 35 wt% if its melt temperature is 10–20 °C above the processing temperature, and 50 wt% API otherwise. These starting compositions are intentionally selected to have a high likelihood of incomplete API solubilization to allow quantification of the dissolved portion using the residual crystalline signal, eliminating the need to prepare multiple mixtures across a range of API loadings: saving time and materials; 50 wt% is not used as a default in all cases, as excessive amounts of undissolved solids can negatively impact the mixing capability and ease of material recovery.

If the achievable API loading is close to the desired loading (specific to each API), we proceed to assessing the API degradation.

The same sample used to predict API loading is assessed for API degradation. While the degradation prescreening serves to generate starting temperatures, the true decision for the appropriate processing temperature range should be decided based on assessment of the MiniMixer sample(s). Mixing can lead to increased degradation, and every API has different requirements for acceptable degradation levels associated with specific related substances. Therefore, an API-specific quantification of degradation (like HPLC) on a sample best mimicking the extrusion process will result in the most reliable way to determine whether extrusion is feasible based on the known requirements for an API’s purity.

These predictions are meant to be a starting point for assessing HME feasibility and selecting initial parameters for HME manufacture on lab-scale extrusion equipment, but not intended to be directly extrapolated to full-scale extrusion. On full-scale extrusion equipment, many factors such as the screw design and feed rate, elevated shear rates (and subsequent temperature increases), and pressure all add complexity to the situation.

The MiniMixer is also not intended to provide information on material viscosity (i.e., processability), although the T_g_ of the resultant material, relative to the pure polymer, provides a sense of the API’s plasticization effect.

Lastly, along with manufacturability, performance and stability are equally critical to the relative success of a given ASD. While manufacturing a fully amorphous extrudate-like product is possible with the MiniMixer, it is not recommended to use this material for performance and stability due to the limited sample and wide particle size range of the ground product. As ASD particle size can have significant impacts on both performance and stability, extrudate material should be sieved to a controlled particle size distribution before testing. Obtaining enough MiniMixer material within a given size range to conduct subsequent performance and stability studies would be challenging. Therefore, we recommend assessing performance (e.g., dissolution/supersaturation) and stability during the process screening phase of our workflow, that is, after determining manufacturability.

#### 4.3.3. Process Screening

If HME is established as a feasible option, process screening is performed to better understand the impacts of time, temperature, and shear to the product quality while generating enough material to perform initial performance and stability studies. The collective results from process screening can subsequently guide extrusion scale-up using established protocols [49].

## 5. Conclusions

A flowchart is presented for screening the feasibility of manufacturing an ASD by HME using less than 200 mg of API. This procedure is aided by the use of a simple device called the MiniMixer, which allows for simultaneous heating and mixing of a sample at HME-relevant temperatures and times. The applied mixing at temperature offers significant advantages over current milligram-scale HME screening approaches, resulting in API loading and degradation predictions closer to what can be achieved in an extruder with minimized experimental time and material burden. This screening process can easily be used to assess the HME manufacturing feasibility of a given API in any extrudable polymer. Thus, implementing this screening process can generate valuable information with regards to HME processing using only sub-gram quantities of API.

## Figures and Tables

**Figure 1 pharmaceutics-16-00076-f001:**
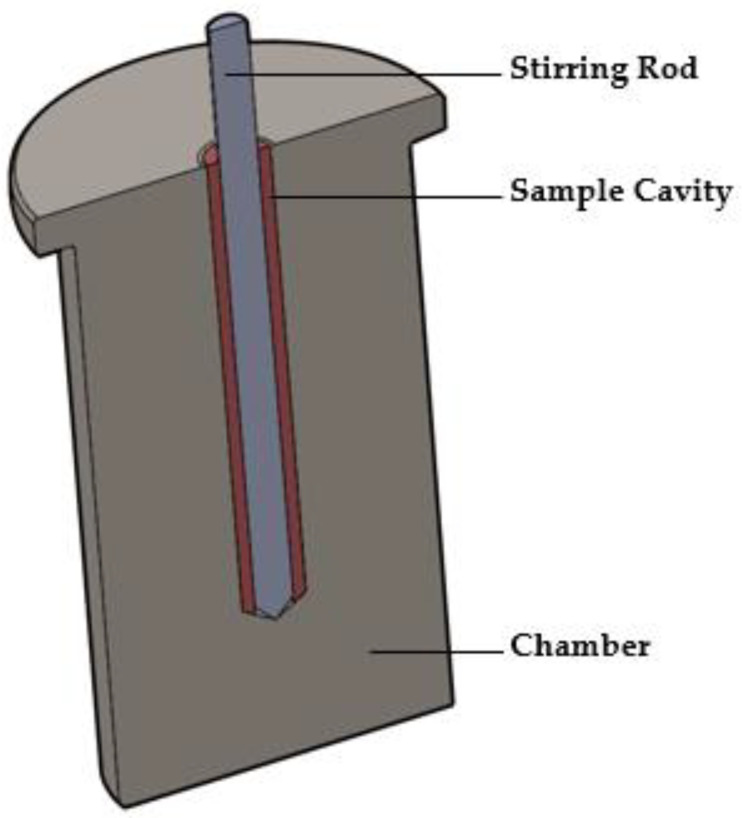
Illustration of the cross section of the MiniMixer.

**Figure 2 pharmaceutics-16-00076-f002:**
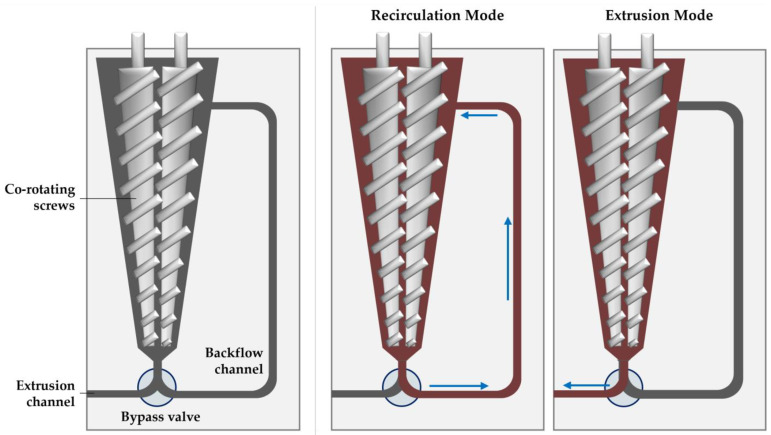
Illustration of the micro compounder (**left**), showing the sample path during the recirculation (**middle**) and extrusion modes (**right**).

**Figure 3 pharmaceutics-16-00076-f003:**
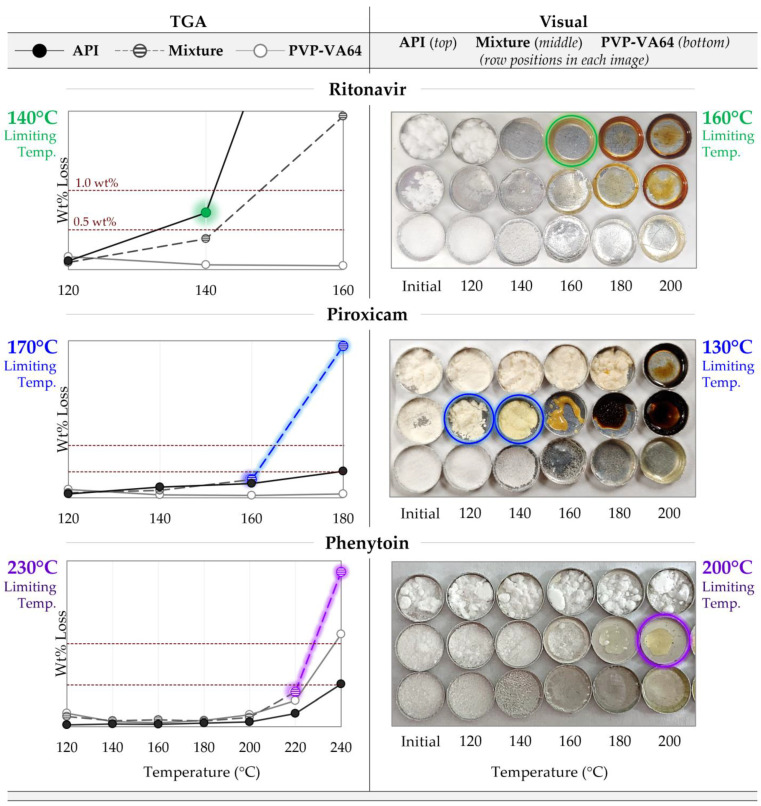
Degradation prescreening results for the three API systems, with the limiting temperatures listed for each test. (**Left**) TGA results showing the wt% mass loss after 15 min at each temperature, with the dashed lines showing the 0.5–1 wt% thresholds. The colored data highlight those that led to the determined limiting temperature for each system. (**Right**) Visual assessment results showing images of the samples after being held isothermally for 15 min at each temperature. The circled samples show the data that led to the determined limiting temperature for each system.

**Figure 4 pharmaceutics-16-00076-f004:**
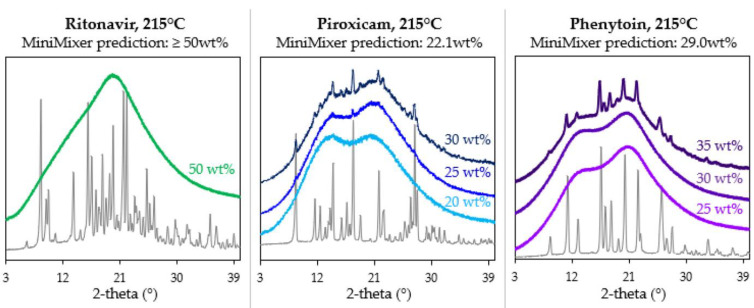
Diffractograms of the samples processed on the micro compounder at various API loadings, with the corresponding API diffractogram shown in gray for each system.

**Figure 5 pharmaceutics-16-00076-f005:**
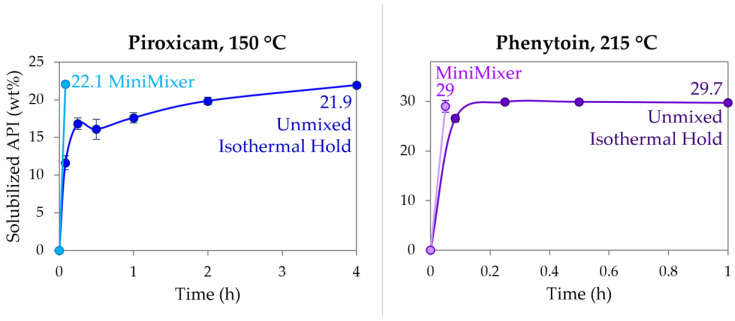
Dissolution of PXCM (**left**) and PHY (**right**) into PVP-VA64 as determined by PXRD at 150 °C and 215 °C, respectively, comparing an unmixed sample and MiniMixer processing.

**Figure 6 pharmaceutics-16-00076-f006:**
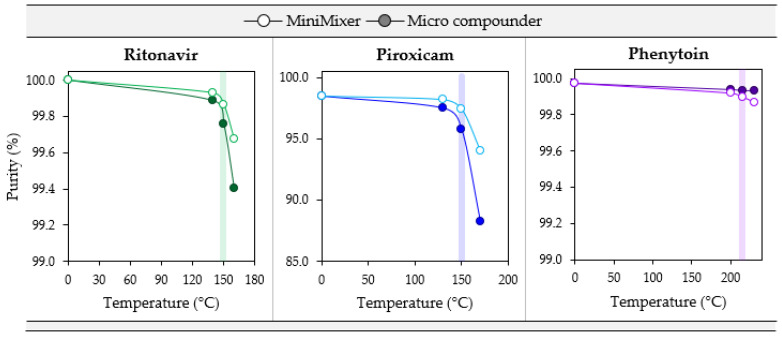
Degradation comparisons for PXCM (20 wt%), RTV (50 wt%), and PHY (25 wt%) MiniMixer (lighter, open circles), and micro compounder processed extrudates (darker, filled circles) as a function of temperature with 3 min of mixing. The colored bar highlights the data collected at the maximum processing temperature determined from degradation prescreening.

**Figure 7 pharmaceutics-16-00076-f007:**
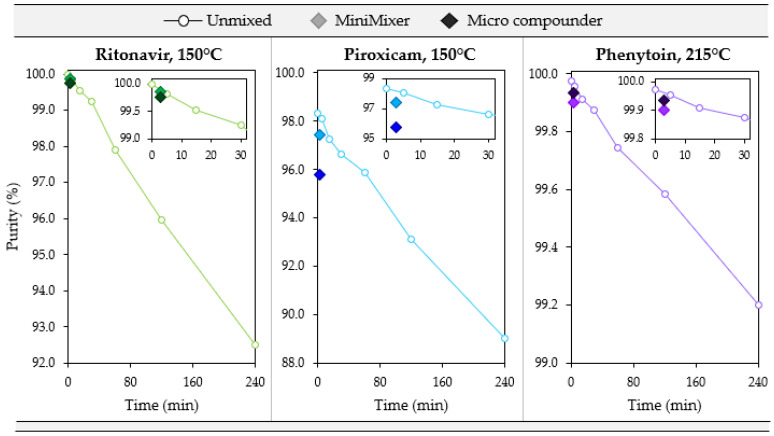
Degradation over 4 h for an unmixed sample (open circles) compared to the 3 min processed MiniMixer (lighter filled diamond) and micro compounder (darker filled diamond) samples for the three systems studied. The inlay shows a close-up of the first 30 min.

**Figure 8 pharmaceutics-16-00076-f008:**
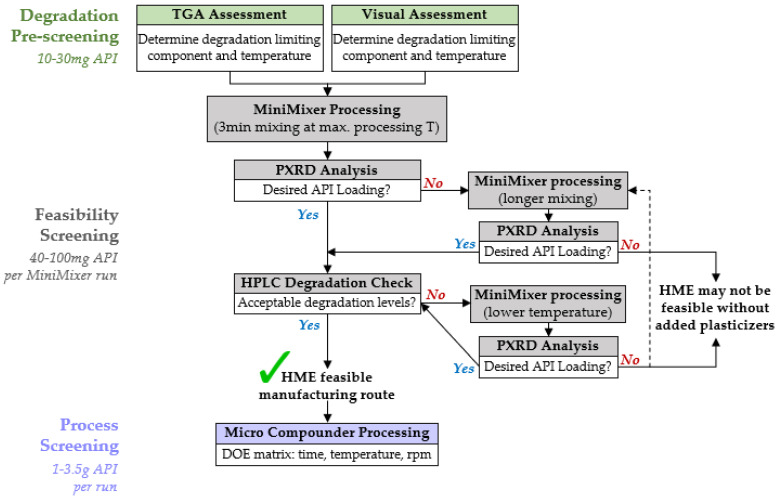
HME feasibility screening flowchart for a given API–polymer formulation.

**Table 1 pharmaceutics-16-00076-t001:** Summary of approaches commonly used for HME screening.

Category	Description	Material Needs	Uses In-Going API Form	HME Relevant Time at Temperature	Applies Mixing or Shear	Ability to Assess Process-Relevant Degradation
Modeling	Flory Huggins, PC-SAFT, solubility parameters, etc.	None-mg	Yes ^1^	No	No	No
Analytical	DSC	mg	Yes	No ^2^	No	No ^4^
	Thermal microscopy	mg	Yes	No ^2^	No	No ^4^
	Rheology	mg—g	Yes	No ^2^	Yes ^3^	No ^4^
	Film casting	mg—g	No	No	No	No
Devices	Miniaturized extruder prototype	mg	No	No	Partially	No
	Vacuum compression molding	mg—g	No	No	No	No
	Micro compounders	g	Yes	Yes	Yes	Yes
	Small-scale extruders	g	Yes	Yes	Yes	Yes

^1^ If the inputs used are experimentally derived. ^2^ While it is possible to run each of these methods with HME-relevant timescales, they are typically conducted at longer timescales and until the sample has reached equilibrium. ^3^ Depending on the method executed and type of rheometer used, the applied shear may not be the same as a material experiences during extrusion. ^4^ In practice, one could recover the material from these tests and assess degradation, but since the processes are not physically relevant, the resultant degradation would not be representative.

**Table 2 pharmaceutics-16-00076-t002:** Summary of model APIs used in this study.

Compound	Structure	BCS Class	Tm (°C)
Ritonavir (RTV)	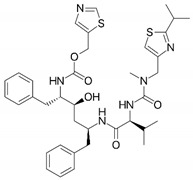	IV [51]	126 [52]
Piroxicam (PXCM)	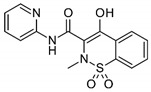	II [53]	198–200 [54]
Phenytoin (PHY)	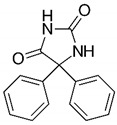	II [53]	295–298 [55]

**Table 3 pharmaceutics-16-00076-t003:** Summary of all MiniMixer and micro compounder runs conducted.

Processing Round	API	Temperature (°C)	API in Ingoing Physical Mixture (wt%)	
			MiniMixer	Micro Compounder
1	RTV	150	50	50
2		140	50	50
2		160	50	50
1	PXCM	150	20	20, 25, 30
2		130	20	20
2		170	20	20
1	PHY	215	25 ^1^, 35 ^1^	25, 30, 35
2		200	25	25
2		230	25	25

^1^ 25 wt% sample was initially prepared to assess API loading, but found to be amorphous once processed. Thus, it was decided to test a 35% loading, which did result in residual crystals for quantification of achievable API loading.

**Table 4 pharmaceutics-16-00076-t004:** Summary of all unmixed isothermal holds prepared.

API	API in Ingoing Physical Mixture (wt%)	Temperature (°C)	Time (h)
RTV	50	150	0.08, 025, 0.5 1, 2, 4
PXCM	25	150	
PHY	35	215	

**Table 5 pharmaceutics-16-00076-t005:** HPLC gradient method for PHY purity analysis.

Time (Min)	Mobile Phase A (%)	Mobile Phase B (%)
0	75	25
10	75	25
30	25	75
40	25	75
40.01	75	25
45	75	25
0	75	25

**Table 6 pharmaceutics-16-00076-t006:** Summary of results from the degradation prescreening.

		RTV		PXCM		PHY	
		TGA	Visual	TGA	Visual	TGA	Visual
Limiting Temperature (°C)	API	140 *	160 *	190	180	240	>220
	Mixture	150	160	170 *	130 *	230 *	200 *
	PVP-VA64	230	200	230	200	230	200
Maximum processing temperature (°C)		150		150		215	
Degradation-limiting component		API		Mixture		Mixture	

* degradation limiting component.

**Table 7 pharmaceutics-16-00076-t007:** Summary of achievable API loadings (wt%) in PVP-VA64 as determined by PXRD.

API	Temperature (°C)	MiniMixer	Micro Compounder
RTV	150	≥50	≥50
PXCM	150	22.1 ± 0.3	24.4 ± 0.2 (25 wt% sample)
			27.6 ± 0.4 (30 wt% sample)
PHY	215	29 ± 1	30.8 ± 0.7 (35 wt% sample)

**Table 8 pharmaceutics-16-00076-t008:** Comparison of the API degradation between the MiniMixer, micro compounder, and 3 min unmixed (prediction as determined from the linear fit). The % error with respect to the micro compounder result is also shown, with negative and positive values indicating purity results that were lower than or higher than the micro compounder, respectively.

API (Starting Purity)	Processing Temperature (°C)	API Purity (3 min)			% Error w.r.t. Micro Compounder	
		Micro Compounder	MiniMixer	Unmixed	MiniMixer	Unmixed
RTV(100)	140	99.89	99.93	—	+0.04	—
	150	99.76	99.87	99.89	+0.11	+0.13
	160	99.40	99.68	—	+0.28	—
PXCM(98.4)	130	97.53	98.22	—	+0.71	—
	150	95.78	97.44	97.94	+1.7	+2.3
	170	88.28	94.00	—	+6.5	—
PHY(99.97)	200	99.94	99.92	—	−0.02	—
	215	99.94	99.90	99.95	−0.04	+0.02
	230	99.94	99.87	—	−0.07	—

## Data Availability

Data are contained within the article or Appendix A.

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
