# Peer review of "Material-Sparing Feasibility Screening for Hot Melt Extrusion"

_pharmaceutics, 2024, doi:10.3390/pharmaceutics16010076_

Round 1

Reviewer 1 Report

Comments and Suggestions for Authors

The manuscript describes a systematic and scientifically up to date proposal to screen API and polymers for amorphous solid dispersion preparation via hot melt extrusion (HME). The novelty is the use of a novel device "mini mixer" with integrated mixing “in-situ”. The manuscript is well written and structured. However, the results could be presented in a more clear way to the first-time reader. I have a few minor comments to address:

Table 2: since the degradation screening was being performed in the paper, it would be interesting to the reader to add info, which kind of degradation products are generated with each molecule – oxidation, hydrolysis, ring opening etc. If applicable, the degradation temperature of each API can be added to the table. It would be also interesting to discuss if the APIs have plasticizing effect and if there is a difference in their glass forming properties. Please also add the API abbreviations either in the table or next to the name, i.e. ritonavir (RTV)

Line 210: please double check if the length is really 37 cm, based on other dimensions I assume it is 37 mm?

Line 257: Micro Compounder is not a “real” hot melt extruder with the classic (twin)screw configuration like conveying and kneading elements etc, barrel shape, die diameter etc. For the reader, I would add a scheme of micro compounder next to the mini mixer for direct (visual) comparison: this way, the results in the manuscript would be much easier to interpret/compare.

Table 3 and 4 look very similar, for direct comparison and convenience to the readers, I would merge them into one table.

Figure 2 is a bit unclear. The circles labelling the chosen samples should be made more bold/visible. It is also not explicit, what do the temperatures in colour mean (assume degradation temperatures for both methods?) For clarity I would also add the descriptions API, mixture and PVP-VA64 on the right sides in each row.

Figure 4 clearly shows the effect of mixing during melting. If particle size distribution (PSD) of the APIs and polymer are available (experimental results or raw material specification) it would be interesting to correlate this data with the results – assume similar PSD of API and polymer leads to more homogenous mixing and faster melting/dissolution of API in the polymer. Nevertheless, I would mention this in discussion in a few words.

Figure 5: The degradation time in the caption text is missing, please add (3 minute, right?)

Lines 703-704: I would add the Tg of PVP-VA64 somewhere (around 110 deg C I think?) so the reader can see the difference between Tg of the polymer and Tm of API and easier interpret the results.

Generally, I would add the importance of screw configuration, i.e conveying and kneading elements in the discussion. In HME, the homogenisation is achieved with screw elements and thus the results obtained in the manuscript may vary in HME. Did the authors consider designing mini screw elements for the mini mixer resembling the conveying and kneading elements?

Author Response

Thank you for your time and effort spent reviewing this manuscript. Your suggestions have helped us better communicate this story. Please see the attached document which details our specific responses to your comments. 

Reviewer 2 Report

Comments and Suggestions for Authors

Dear authors,

Thanks for submitting this interesting article and giving me the chance to comment on it in the reviewing process. Overall, it provides an approach to screen API-polymer-mixtures with respect to their manufacturability via hot melt extrusion in a material sparing way. 

I would like to comment on a few topics and hope that they support you in adjusting your manuscript.

  1. Your screening approach is very useful in order to provide some guidance on the manufacturability of a certain API-Polymer-Mixture and selecting suitable manufacturing parameters for the small scale extrusion experiment. However, it does not address topics such as the dissolution/supersaturation potential and or physical stability aspects, which are both important aspects to be considered during an ASD screening → Please add a few sentences so that the reader can better follow your thoughts and understands in which phase of an ASD screening your workflow is best placed, e.g. would you assess the dissolution/supersaturation potential before or after; how and when would you assess the physical stability of the system. 

  2. Despite the fact that microcompounders are widely used in literature as a screening tool they are somewhat limited with respect to the shear rates they can generate compared to a standard extruder. This difference in the shear rate sometimes leads to the fact that a certain physical mixture cannot be processed on a MicroCompounder into an ASD, but is easily transformed into an ASD on a standard extruder. → Please clearly state in your article that the shear rates in your MiniExtruder are comparable to the shear rates in a Micro Compounder, but that they are lower compared to an extruder so that dispensing on the systems some false negative results (e.g. screening says the system cannot be manufactured, but due to the higher shear rates in a standard extruder it could be manufactured) can occur. 

  3. With this article you introduce the MiniExtruder as a new screening tool. While limiting the article to one polymer and three different APIs is justified (to allow the reader to follow the results better), you should add the following information:

    1. How easy can the samples be removed from the MiniExtruder for the described polymer Copovidone?

    2. Which other polymers have you tested so far with the device, how easy could you remove samples with other polymers from the device and are the results comparable?

  4. You propose two different methods for the degradation pre-screening: a TGA based weight loss and an assessment of discoloration after thermal treatment. You nicely mention the limitations of both methods and I was wondering why you are not proposing to use an HPLC method similar to the tests you do with the MiniExtruder samples.

  5. Figure 7: The term “Desired API loading?” might be misleading. Based on the text description I would think that you check here the amount of API that can be dissolved in the polymer, or? → Consider rephrasing to make it more clear.

Author Response

(The authors gave the same response as above.)
